# Autonomic Nerve Activation Observed for Hemodialysis Patients While Squeezing a Soft Ball

**Jian-Chiun Liou [1],\*, Chih-Wei Peng [1] , Philippe Basset [2] and Zhen-Xi Chen [1]**

[1] School of Biomedical Engineering, Taipei Medical University, Taipei 11031, Taiwan; cwpeng@tmu.edu.tw (C.-W.P.); hoppyshi@gmail.com (Z.-X.C.)
[2] ESYCOM, Université Gustave Eiffel, CNRS, CNAM, ESIEE Paris, F-77454 Marne-la-Vallée, France; philippe.basset@esiee.fr
\* Correspondence: jcliou@tmu.edu.tw

**Abstract:** In this study, a medical grade pulse rate (PR) instrument was used to monitor hemodialysis patients, and the wearable product was applied for the 4 h observation. Electrocardiogram (ECG) and photoplethysmography (PPG) data were simultaneously collected to observe physiological phenomena in patients undergoing hemodialysis. The analyzed results of 38 patients undergoing the treatment (as sympathetic/parasympathetic balance indicators before-hemodialysis (HD), and after-HD) and autonomic nerve activation for the pulse rate (PR) measurement accompanied by squeezing a soft ball were also observed. The results prove the pulse rate measurement while squeezing the soft ball and analyze data, and we show that the analyzed results have a very concentrated normal distribution. This study presents oxygen saturation (SpO$_2$) and continuous pulse rate distribution curves during the 4 h observation of the hemodialysis patients and we show that some patients undergoing kidney dialysis have sleep apnea. They become lethargic during dialysis and experience severe hypoxia due to intermittent respiratory arrest. Studies have confirmed that such monitoring and biofeedback designs can reduce the incidence of hypotension during dialysis.

**Keywords:** pulse rate (PR); hemodialysis; oxygen saturation

## 1. Introduction

Intradialytic hypotension (IDH) is a common complication during hemodialysis treatment, about 20–30% patients experience this condition during the treatment process. IDH often makes it difficult for patients to return to their original weight and also causes the reduction in dialysis adequacy as well as numerous cardiovascular disease events, thereby increasing patients' mortality. The etiology and theory are as follows:

A. Excessive reduction in effective blood volume: (1) if the ultrafiltration rate is too fast during the hemodialysis treatment process, the amount of ultrafiltration is too great, which will make the ultrafiltration rate greater than the liquid in the surrounding tissue gap. If the blood vessel lumen refills too slowly, there is insufficient effective blood volume, and this will eventually cause the patients' hypotension. (2) The target and weight are set too low: when ultrafiltration occurs in the patients and they experience weight loss during the hemodialysis treatment process, the rate at which the blood vessel cavity refills, in terms of the fluid and the surrounding tissues, slows down, and some patients experience little or no weight loss during the dialysis interval. The increased risk of first aid indicates that when the body does not have too much fluid retention, excessive fluid removal during dialysis may cause hypotension during or after dialysis. (3) When the sodium coating of the dialysate is lower than the sodium coating of the plasma, the osmotic pressure of the patients' plasma drops, and the water

in the blood enters the cells extracellularly, which causes a sharp decrease in effective blood volume. This is the main reason for an early drop in blood pressure during hemodialysis treatment [1–6].

B. Vasomotor dysfunction: the decrease in blood volume will cause a decrease in cardiac output due to the reduced heart filling. When the filling of the heart decreases, the increase in heart rate has little effect on cardiac output. Since more than 80% of blood volume is in the veins, an increase in venous volume causes the decrease in heart filling and cardiac output to result in hypotension, and even the visceral and subcutaneous vascular bed volumes have great capacity for variation. At present, it is thought that most of the changes in venous volume are due to passive expansion of the venous bed and retention of blood in the vein because of the pressure transmitted by the distal arterioles. This is not important for patients with vasodilators and large blood volumes—a condition that often occurs in kidney disease. As the kidneys are dysfunctional, the water and sodium in the body cannot be excreted, which causes water and sodium retention.

If the heart is filling more than normal, patients who have low blood volume and the increase in blood retention in the veins can experience hypotension. In addition, the degree of arteriolar contraction and the total peripheral resistance (TPR) are also important, because TPR has a decisive effect on blood pressure at any cardiac output level [7–15]. The factors that cause abnormal blood pressure include: (1) acetic acid dialysate: this can easily cause cardiac myocardial depression, vasodilation, and IDH. (2) Dialysate overheating: the dialysate temperature has a strong stimulating effect on vasodilation, which can lead to dilation of the veins and arteries. (3) Eating during dialysis: eating can cause visceral blood vessels to dilate, leading to a decrease in TPR and an increase in visceral venous volume. The "food effect" on blood pressure lasts at least 2 h. (4) Tissue ischemia: this can cause adenosine release, and then adenosine can block the release of norepinephrine and cause endogenous vasodilation. Severe hypotension can amplify its own effects through the following mechanisms: hypotension, ischemia, adenosine release, norepinephrine release, and vasodilation. Clinical observations have shown that patients with low hematocrit (less than 0.20–0.25) are more likely to develop dialysis hypotension. (5) Autonomic neuropathy: autonomic neuropathy is more common in patients with diabetes. When the blood volume of such patients is reduced, the arterial contractility is also impaired. As a result of the decrease in cardiac output, the patients' ability to maintain blood pressure also decreases. For patients that are prone to experience the problems of hypotension, even in the absence of autonomic neuropathy, the response to plasma norepinephrine during hypotension is usually lower.

C. Cardiac factors: (1) diastolic dysfunction: patients have asymmetric ventricular hypertrophy and left ventricular outflow tract disorder, which will lead to a decrease in the patients' heart-filling volume. (2) Heart rates and cardiac contractility: when the ejection volume of the heart is not limited by the filling volume, the heart can compensate for the decrease in TPR by increasing the heart rate and contractile force to increase the cardiac output [16–22]. When cardiac compensation mechanisms are impaired by other reasons, for example patients have uremic autonomic neuropathy or take ß blockers, or they are elderly patients, the heart rate cannot increase accordingly and a slight decrease in TPR can lead to hypotension. (3) Decrease in cardiac output caused by other reasons: if patients are of an old age, or experience hypertension, periarterial sclerosis, myocardial calcification, valvular disease, or amyloidosis, their myocardial contractility will be weakened. Other uncommon reasons include pericardial tamponade, myocardial infarction, potential bleeding, sepsis, abnormal heart rates, response of the dialyzer, hemolysis, and air embolism.

Renal failure parameters are determined in the clinic, renal failure is a clinical condition when the kidney cells are damaged for some reason and cannot effectively remove metabolic waste and water from the body. The rate of hematocrit change, total $CO_2$ ($tCO_2$) and urea per unit change of creatinine during early renal failure (RF) (creatinine less than 5 mg/dL) was significantly higher than during moderate (creatinine between 5 and 10 mg/dL) or advanced (creatinine greater than 10 mg/dL) RF. If this condition is chronic and irreversible, it is called chronic renal failure. When the kidney function declines to a serious level and there are extensive systemic symptoms, it is called uremia [23].

Acute renal failure (ARF) refers to a group of clinical manifestations in which renal function declines rapidly in a short period of time due to the kidney itself or external factors, manifested as azotemia, water and electrolyte disorders, metabolic acidosis, anemia, and bleeding inclination etc. Acute risk factors include volume depletion, aminoglycoside, radiocontrast exposure, septic shock, hypotension, and pigmenturia. Chronic risk factors include pre-existing renal disease, hypertension, congestive heart failure, diabetes (DM), age and liver cirrhosis.

The role of the kidneys in the regulation of arterial blood pressure (mainly the renin-angiotensin-aldosterone axis), renin-angiotensin system (RAS) or renin-angiotensin-aldosterone system (RAAS) constitutes a hormone system. When a lot of blood loss or blood pressure drops, this system will be activated to help regulate the body's long-term blood pressure and extracellular fluid volume (body fluid balance).

When blood pressure drops, the kidneys secrete renin. Renin catalyzes the hydrolysis of angiotensinogen to produce angiotensin I. Angiotensin I basically has no biological activity but is cleaved by the two amino acid residues at the C-terminus by the angiotensin converting enzyme (ACE) to form angiotensin II. Angiotensin II is highly effective on the contraction of blood vessels, thereby increasing blood pressure. Angiotensin II can also stimulate the adrenal cortex to secrete aldosterone. Aldosterone can promote the reabsorption of water and sodium ions by the kidneys, which in turn increases fluid volume and raises blood pressure.

The aim of the study is to use a system to simultaneously observe the pulse rate (PR)/SpO$_2$ data of hemodialysis patients for 4 h, which could be used for microscopic examination of the patients' micro-physiological signals during the entire hemodialysis process. Cardiac sympathetic response (CSR) and malnutrition-inflammation syndrome (MIS) with a score from the squeeze test with a soft ball are validated assessment tools for patients' health condition. We aim to evaluate the joint effect of CSR and MIS on all-cause and cardiovascular (CV) mortality in patients with hemodialysis (HD). The methods and SpO$_2$ analysis/pulse rate (PR) analysis also have a stable effect during hemodialysis treatment. Changes in normalized low frequencies ($\Delta$nLF) during HD were utilized for quantification of CSR.

The kidneys can metabolize substances harmful to the human body and play an important role in maintaining life. If you suffer from diabetes, kidney stones, renal arteriosclerosis and other diseases and are not properly treated, and the kidney is damaged for more than three months, it can cause chronic kidney disease, and then damage the kidney function. Therefore, such patients must receive hemodialysis treatment.

Hemodialysis involves sticking one end of a puncture needle into the distal end of the arteriovenous fistula. It drains the blood out of the body, removes urinary toxins and water through the diffusion and ultrafiltration of the artificial kidney's (kidney dialysis machine) semipermeable membrane, and then leads the blood back to the proximal end of the arteriovenous fistula. The duration of kidney dialysis is about 4 h, and the treatment happens three times a week.

Hemodialysis is through the diffusion function (diffusion), so that toxins are discharged through different concentrations. The filtration capacity is less than 0.5–1.0 L/h, and the molecules that can be removed are small. The molecular weight of uremic molecules removed by hemodialysis is about 1000 or less. Hemofiltration gives the blood a positive hydrostatic pressure, the dialysate produces negative hydrostatic pressure and convection (convection), and toxins are filtered out through a semipermeable membrane. The filtration volume of hemofiltration is >1.0–2.0 L/h, the filterable molecular weight can reach 25,000, but the small molecules are not clean. The percentage of subjects receiving hemodialysis treatment every day is 1/5.

## 2. System Architecture for Capturing Physiological Signals

Cardiac function can reflect much information about the human body, including health status, daily lifestyle, emotional states, and the early onset of abnormal heart activities. In traditional medical testing equipment, the wave rates of heartbeat signals and the flow activities of cardiac blood are monitored by measuring electrophysiological signals and using an electrocardiogram (ECG) [24–30].

These signals require connecting electrodes with the patients' bodies to measure the electrical activities of the cardiac tissues. In addition, as the heart beats, a pressure wave is transmitted through the blood vessels, which slightly changes their diameter.

In addition to ECG, another option is photoplethysmography (PPG), an optical technology that can obtain blood oxygenation information related to cardiac blood flow function without measuring bioelectrical signal transduction. PPG is mainly used to monitor and measure the blood oxygen distributed saturation signals ($SpO_2$) in the blood. It can provide the signal output information of cardio-bioelectrical function without measuring bioelectrical signal transduction. Through PPG technology, a heart rate monitor can integrate finger blood oxygen PPG measurements and extend them to portable display devices, such as watches or wristbands, to achieve real-time detection applications, as shown in Figure 1. Generally, the high-precision $SpO_2$ value and pulse rate architectures are the solutions that combine analog front-ends and analog-to-digital converter singles (ADC). This can be used to monitor the blood oxygen content of body tissues.

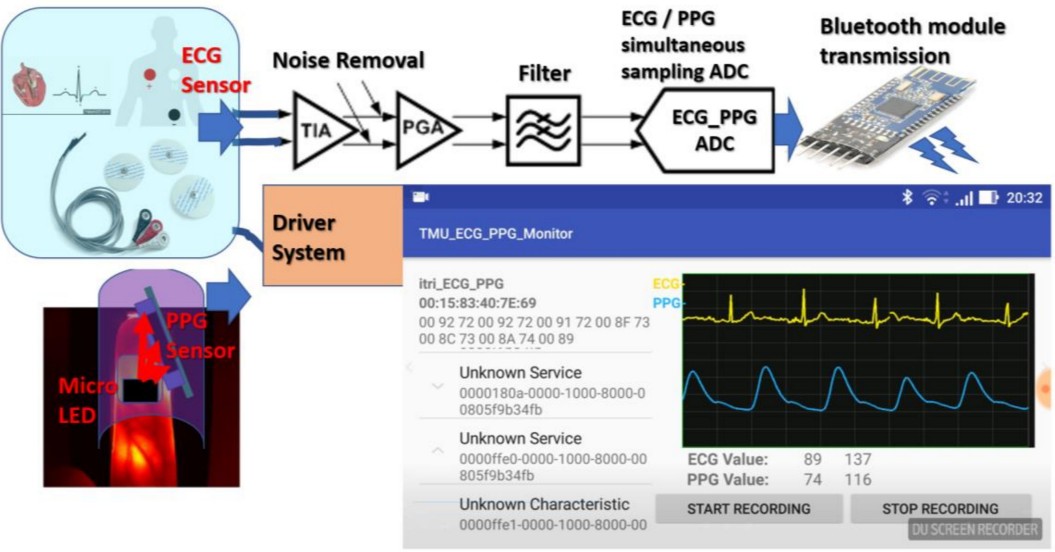

**Figure 1.** System architecture for capturing physiological signals.

The body's intake and utilization of oxygen is a complex biological process. Generally speaking, to determine the status of tissues obtaining and using oxygen, two factors must be tested: the tissues' oxygen supply and the tissues' oxygen consumption. Measuring blood oxygen parameters is necessary to understand the body's oxygen acquisition and consumption:

(A) Partial pressure of oxygen ($PO_2$)

Tension created by oxygen is physically dissolved in the blood. The arterial blood oxygen partial pressure ($PaO_2$) level mainly depends on the oxygen partial pressure of the inhaled gas and external respiratory function, and it is also a dynamic factor for the diffusion of oxygen into the tissue; $PvO_2$ reflects the state of internal respiratory function.

(B) Oxygen saturation ($SO_2$)

$SO_2$ is defined as the percentage of Hb bound to oxygen.

$$SO_2 = (\text{oxygen content} - \text{physically dissolved oxygen}) \div \text{oxygen capacity} \times 100\%$$

This value is mainly affected by $PO_2$, and there is a relationship between the dissociation curve of oxygenated Hb. Normal arterial oxygen saturation is 93–98% and venous oxygen saturation is 70–75% [31].

The block diagram of SpO$_2$ analysis using the equipment of the pulse rate (PR) analyzer is shown in Figure 2, and it clinically captures the physiological analysis signals of SpO$_2$ from the PR analyzer by the external heart rate acquisition system. The next stage is the observation of autonomic nerve activation, which analyzes and collects the signals of heart rate variability HRV after hemodialysis treatment. The third stage is to obtain the signals of SpO$_2$ from the PR analyzer. These patients undergoing hemodialysis treatment have different complications. As the hemodialysis treatments are carrying on, the observations on to which extent the HRV affects the sympathetic and parasympathetic nerves are also carrying on.

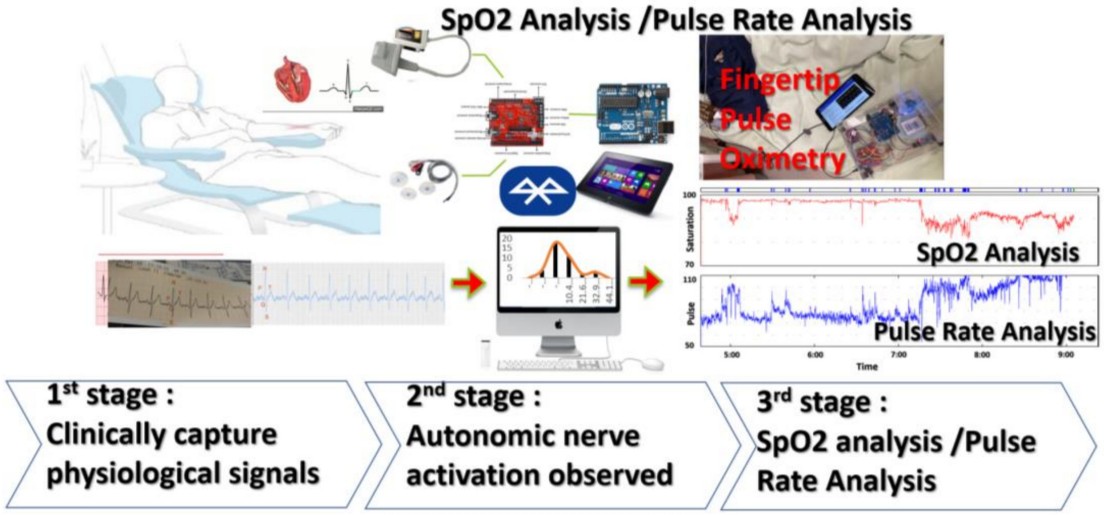

**Figure 2.** The block diagram of oxygen saturation (SPO$_2$) analysis using the equipment of pulse rate analysis.

The work performed concerns the main research areas of experimental focus:

(1) For hemodialysis patients, the high-precision SpO$_2$ values and pulse rate architectures are the solutions that combine the analog front-ends and analog-to-digital converter singles (ADC).

(2) For patients undergoing hemodialysis, clinically capture the physiological analysis signals of SPO2 from the PR analyzer with the external heart rate acquisition system.

(3) For mortality risk assessment of hemodialysis patients, observee the autonomic nerve activation, which analyzes and collects the signals of HRV after hemodialysis treatment.

(4) The next stage is to obtain the SpO$_2$ signal from the PR analyzer. These patients undergoing hemodialysis treatment have different complications. For this reason, with the progress of hemodialysis treatment, observations on the degree of influence of HRV on sympathetic and parasympathetic nerves are also underway.

(5) ECG and PPG can be detected simultaneously to observe physiological phenomena in patients undergoing hemodialysis. The ECG/PPG signals can be synchronously sampled by using ADC, and synchronous counting architecture simultaneously drives sampling blood oxygen and heart rate detection to achieve the applications of both continuous detections.

(6) SpO$_2$ analysis/pulse rate (PR) analysis also has a stable effect during hemodialysis treatment. Changes in normalized low frequencies ($\Delta$nLF) during HD were utilized for quantification of CSR.

An ECG is a periodic waveform, which includes a P waveform signal, a QRS (A combination of the Q wave, R wave and S wave, the "QRS complex" represents ventricular depolarization.) waveform group signal, and a T waveform signal. P waveform signals represent the changes in atrial systolic potential, QRS wave groups represent the changes in ventricular systolic potential, and T waves represent the changes in ventricular diastolic potential. The measurements or evaluations of the

heartbeat rates are represented by the interval between the periodic signals. Where R is the point corresponding to the PQRS (the P wave followed by the QRS complex and the T wave) peak of the heart rate ECG wave, R-R is the interval between consecutive RS (RS-type complex in lead with a steady increase in the relative size of the R-wave toward the left chest and a decrease in the S wave amplitude). When a light beam with a special wavelength is incident on the skin surface of a finger, each time a heartbeat occurs, the contraction and expansion of the blood vessel affects the transmission of this light beam. The light beam can pass through the fingertip, and when it passes unobstructed through blood vessel tissue and reflects onto a photosensitive sensor, the light beam is attenuated to a certain extent. The absorption of light by veins and other tissues anywhere in the body remains essentially unchanged. It is assumed that the photosensitive sensor stays still when the light beam is passed through. Due to the pulse of light beams that exist in the body's arteries, the absorption of light beams with special light wavelengths is natural and changes. This principle framework converts the intensity of the light energy into a signal transmitted by radio waves. The arteries cause the changes in the absorption of the PPG light beam, whereas the intensity of the energy absorbed by other tissues remains essentially unchanged. The generated radio wave signals can be divided into direct current (DC) and DC wave signals and alternating current (AC) and AC wave signals. Extractions of the AC signals enable the measurements of blood flow characteristics.

The instant values of heart rate can reflect a person's current heart activity, which is a measurement of the body's overall health. Heart rate measurement in hospitals is mostly done by ECG, but this is inconvenient for use in daily activities and sports. As the PPG method is simple and the measuring device is easy to wear, it has gradually become the main method for measuring blood oxygen, pulses, and heart rates under non-hospital conditions. The optical sensors of heart rates can generate a PPG waveform for measuring heart rates and use the data as a basic biometric value. Many other values can be obtained using PPG, although they are less easy to obtain and have lower accuracy. High-quality PPG signals are necessary to gather a large number of biometrics, for which a demand exists.

PPG provides the functions of upgrading software, new features, and enhancements and improvements to the performance of signal captures of ECG and PPG devices. This will be updated with upgraded software and provide detailed information on the enhancement of system interface operation functions. The wireless oscilloscope for the acquisitions of the ECG and PPG signals is a portable mobile phone or a tablet device running Android software. It uses an Arduino Leonardo microcontroller to perform analog acquisitions of digital data. The digital data are transmitted to a portable device interface connected to a Bluetooth device with a flat display, and this device can monitor ECG and PPG signals at the same time. The sensor collects weak ECG and PPG signals and converts them through the ADC of the Arduino Leonardo. At the same time, a training number is transmitted through the HM-10 Bluetooth module. Users can receive and monitor ECG and PPG signals using an app on an Android device (mobile phone or tablet).

From the signal strengths of the PPG and ECG and the individual physiological characteristics of the frequency cycle, we can determine that the ECG peak is caused by a special contraction of the ventricle, and the characteristic PPG peak is caused by vasoconstriction. The period of these two signals is the same, but there is a time difference between the two volumes, as Figure 3 shows. Therefore, we can conduct blood characteristically from the cardio bioelectrical function signal to the measurement site. Time, also known as pulse period transit time (PTT), the cardio output speed at which the travels of pulse period waves are echoing blood pressure corresponds directly to correlation. If the blood pressure is high, the pulse period waveform spreads quickly; otherwise, if it is slow, the ECG signals and pulse wave PPG signals obtain the PTT value, and some conventional body parameters are added using the established characteristic equation. It is a noninvasive continuous measurement observation of the instant blood pressure cycle waveform.

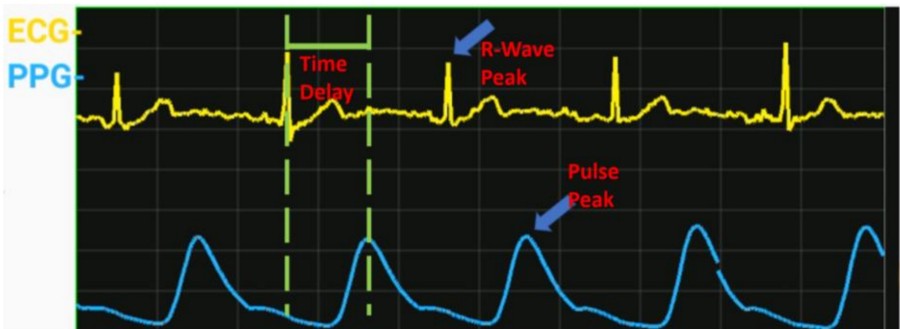

**Figure 3.** Individual physiological characteristic points for photoplethysmography (PPG) and electrocardiogram (ECG).

## 3. Experiment and Results

Hemodialysis (HD) operates through the diffusion function (diffusion), the toxins are discharged through the concentration difference, and the filtration volume is <0.5–1.0 L/h. The molecules that can be removed are smaller. The molecular weight of uremic molecules removed by hemodialysis is about 1000 or less. Hemofiltration (hemofiltration) gives the blood a positive hydrostatic pressure. The dialysate produces a negative hydrostatic pressure and convection (convection), and toxins are filtered out through a semi-permeable membrane. The filtration capacity of hemofiltration is >1.0–2.0 L/h. The filterable molecular weight can reach 25,000, but the small molecules are not clean. It can also be described that hemodiafiltration has both the diffusion effect of hemodialysis (good removal efficiency for small molecules) and the convection effect of blood filtration (a certain removal rate for large, medium, and small molecules). It can be said to be a treatment that combines the best of both. Compared with hemodialysis (HD), hemodia-filtration HDF can increase the clearance of small molecules, and for medium and large molecules that cannot be removed by normal hemodialysis. The clearance rate is n times. It can be used to prevent and ameliorate diseases caused by the deposition of large molecules such as amyloidosis or urotoxic neuropathy.

In continuous venovenous hemofiltration (CVVH), CVVH, a double-lumen catheter is inserted in the vein. The pump is used to pull the blood out through the dialyzer and then returned to the body to remove excess water and toxins. It is necessary to put a double-lumen catheter into the large vein and slowly remove excess water and urinary toxins from the body with a low blood flow rate (about 150–200 mL/min) without interruption for 24 h. As conventional hemodialysis cannot meet the medical needs, the performance of HD is only equivalent to that of a native kidney in stage 5 chronic kidney disease. In addition, hemodialysis is not effective in removing medium and large molecules of urinary toxins, and complications such as hypotension often occur during the process. In addition, the purity of traditional dialysate is insufficient, and inflammation is prone to occur. Therefore, there is continuous venous hemofiltration (CVVH). Regarding dialysis methods (HD, HDF, CVVH) as shown in Table 1, the indications of treatment are different, the principles are slightly different, and the treatment time is also different. The more severe the patient's condition, the more CVVH is used. In the CVVH mode, dialysis is performed for 24 h (−2.0 kg/24 h). For the HD or HDF mode, it is −2.0 kg/4 h.

HD vintage was defined as the duration of time between the first day of HD treatment and the first day that the patient entered the study cohort. Each HD session was performed for 3.5–4.5 h with a blood flow rate of 200–300 mL/min and dialysate flow rate of 500 mL/min. Blood pressure was recorded in the horizontal recumbent position before the dialysis session. Pre-dialysis blood samples were obtained from the existing vascular access.

**Table 1.** Hemodialysis methods (hemodialysis (HD), Hemodia-filtration HDF, continuous venovenous hemofiltration (CVVH)) and characteristics and dialysis time.

| Hemodialysis Methods (HD, HDF, CVVH) | Characteristics and Dialysis Time |
|---|---|
| HemodialysisB (HD) | (A) HD performance is only equivalent to the Native Kidnet of Stage 5 CKD. <br> (B) HD is not effective in removing medium and large molecules of urinary toxins. <br> (C) Complications such as hypotension often occur during HD. <br> (D) Filter capacity <0.5–1.0 L/h. |
| Hemofiltration | (A) The MW of blood filtration can reach 25,000, but the small molecules are not clean. <br> (B) Better cardiovascular stability, less prone to hypotension. <br> (C)Filter capacity >1.0–2.0 L/h. |
| Hemodia-filtration (HDF) | (A) Less blood pressure drop. <br> (B) Could reduce musculoskeletal symptoms caused by long-term dialysis treatment. <br> (C) Improve nerve conduction velocity. <br> (D) Good macromolecular uremic (ß2-Microglobulin) clearance rate. <br> (E) Good small molecule clearance rate (URR). <br> (F) Hemodiafiltration (HDF) = Hemodialysis (HD) + Hemofiltration (HF) |
| CVVH (continuous venovenous hemofiltration (CVVH)) | (A) 24 h without interruption, low blood flow rate (about 150–200 mL/min). <br> (B) Slowly remove excess water and urinary toxins from the body. <br> (C) In CVVH, the double-lumen catheter is inserted into the vein, and the blood is pulled out through the dialyzer and returned to the body by the pump to achieve the purpose of removing excess water and toxins. |

## 3.1. ECG and PPG Signal-Capture Device and HMI Operation

The master–slave integrated module has three functions: transparent transmission, remote control, and the programmable input–output (PIO) pins collection. It is switched and set through an AT ("AT" meaning 'attention') instruction set (the commands for controlling the Bluetooth module are collectively referred as AT commands). This is the same as the Bluetooth serial port module described earlier, with no changes to the printed circuit board (PCB) or lower-level program. The ECG and PPG waveforms and values are displayed on the mobile panel (for long-term recording), as shown in Figure 4.

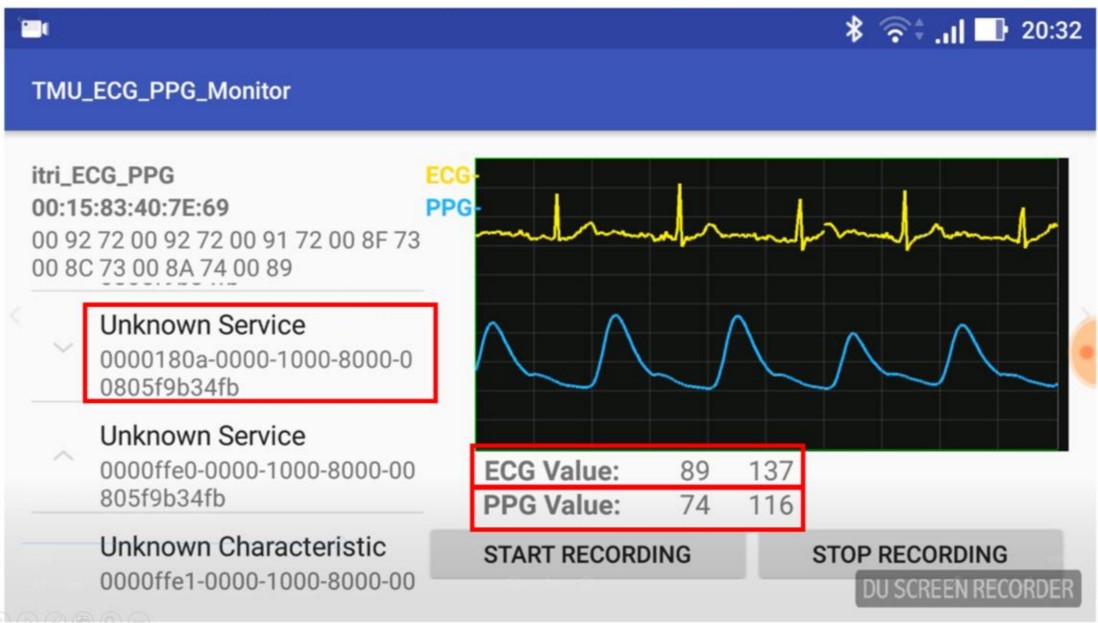

**Figure 4.** ECG and PPG waveforms and values displayed on a mobile panel (for long-term recording).

The capture devices of ECG and PPG signals and human machine interface (HMI) operations consist of the following five main parts, as Figure 5 shows. We designed them to monitor the physiological status of hemodialysis patients during dialysis.

(a) Battery module and power switch.
(b) Arduino Leonardo microcontroller.
(c) HM-10 Bluetooth 4.0 module (BLE).
(d) ECG module (AD8232 Heart Monitor).
(e) PPG module (Easy Pulse V1.1).

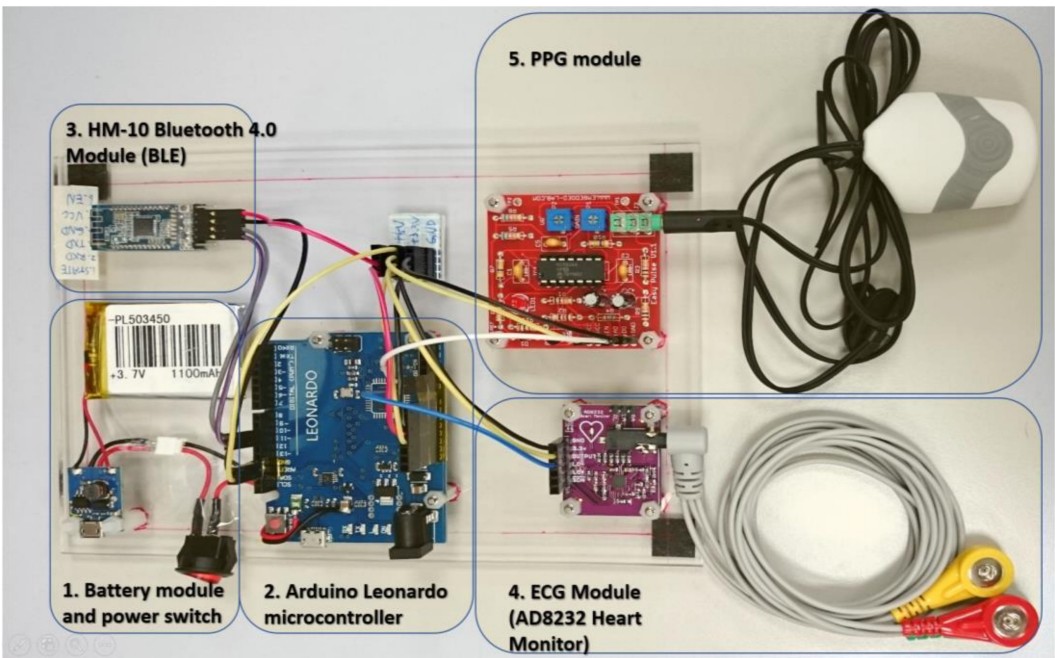

**Figure 5.** ECG and PPG signal capture device and HMI operation.

The design of $SpO_2$ and heart rate during the hemodialysis process for 4 h could be observed simultaneously. The ECG/PPG signals can be synchronously sampled by using ADC, and synchronous counting architecture simultaneously drives sampling blood oxygen and heart rate detection to achieve the applications of both continuous detections. The module is based on the technology of Bluetooth 4.0 BLE, and the core chip used is CC2541 from Texas Instruments (TI). It is mainly used in consumer medical electronic devices, personal sports equipment (such as running shoes and sports watches), personal health maintenance applications (such as heart rate belts), leisure, gaming, human-interface device (HID), remote control equipment, mobile computer peripherals (such as a wireless mouse), and security or other space-sensing short-range applications. It can achieve the effect of ultra-low-power short-range wireless connection under continuous operation using a button battery for more than a year. Both Microsoft and Apple have announced support for the new technology in their upcoming Windows 8 and end products, which means that the technology will be widely used in standard computing and communication platforms.

### 3.2. Capturing Pulse Rate (PR)/$SpO_2$ during Dialysis

Simultaneous acquisitions of pulse rate (PR)/$SpO_2$ data can be achieved for 4 h while patients are receiving the hemodialysis treatment. Blood oxygen content is directly related to the amount of hemoglobin and its saturation with oxygen. In the arterial blood of healthy people, the oxyheme saturation is 98% and that of venous blood is 70%. The dialysis patients have normal oxidized



hemoglobin saturation, but because of the problem of anemia, heme saturation will change from normal 15 g% to 10 g%, so their blood oxygen content is much lower than in healthy people.

Measurement range of $SpO_2$ was in the range of 0~100%, that of pulse frequency was in the range of ~30–230 bpm, accuracy of $SpO_2$ was ±2% (70–100%, one standard deviation), accuracy of the pulse rate was ±2 bpm (range: 50–110) or ±2% of the value (range: 100–230). The system integrates patients' physiological signals during 4 h of hemodialysis treatment and heart rate (ECG) monitoring. We demonstrated that a 4 h simultaneous observation system of ECG/PPG in hemodialysis-treated patients could be used for online observation of physiological signals during the patients' entire hemodialysis-treated process.

We perform the clinical investigation on 38 emergency patients using a $SpO_2$ sensor as shown in Table 2. These patients had problems of increasing heart rate or increasing risk within 4 h of hemodialysis treatment in the hospital. The symptom of the proposed observation method and computer program was the information system. The obtained differences in these values after hemodialysis and before hemodialysis were calculated and compared.

**Table 2.** Summary table (main clinical features).

| Demographic and Clinical Information's | Patients ($n = 38$) |
|---|---|
| Age, years, mean ± SD (hemodialysis treatment) | 68 ± 12 |
| Gender, n (%) | |
| Male (hemodialysis treatment) | 47.4% |
| Female (hemodialysis treatment) | 52.6% |
| The percentage of subjects receiving hemodialysis treatment every day | 20% |
| The filtration volume of Hemofiltration | >1.0–2.0 L/h |
| ΔnLF with sympathetic/parasympathetic balance indicator (value equal to 0.811) | 18/38 |
| ΔnLF with autonomic dysfunction positive (value equal to 10.462) | 11/38 |
| ΔnLF with autonomic dysfunction negative (value equal to −11.987) | 4/38 |
| ΔnLF with autonomic nervous disorders | 5/38 |
| Observing patient issues | |
| With Poorly controlled hypertension ratio | 32/38 |
| With Aortic aneurysm ratio | 16/38 |
| With The blood vessel of bypass surgery ratio | 28/38 |
| The squeeze test with the soft ball | 38/38 |

Poorly controlled hypertension is responsible for the occurrence of stroke, this also means patients who have stroke symptoms. An aortic aneurysm is an enlargement (dilatation) of the aorta to greater than 1.5 times normal size. They usually cause no symptoms except when ruptured. Occasionally, there may be abdominal, back, or leg pain. Bypass surgery can be used for patients with multiple stenosis or blockage of the blood vessel. The above patients with these symptoms are clinically diagnosed.

The recruitment section and clinical diagnostic criteria background of this study consisted of patients with chronic diseases. The aim of this study is using a system to observe the pulse rate (PR)/$SpO_2$ data of hemodialysis patients for 4 h, which could simultaneously be used for microscopic examination of the patients' micro-physiological signals during the entire hemodialysis process. Cardiac sympathetic response (CSR) and malnutrition-inflammation syndrome (MIS) with a score from the squeeze test with a soft ball are validated assessment tools for patients' health condition. We aim to evaluate the joint effect of CSR and MIS on all-cause and cardiovascular (CV) mortality in patients with hemodialysis (HD). The methods and $SpO_2$ analysis/pulse rate (PR) analysis also have a stable effect during hemodialysis treatment. Changes in normalized low frequencies (ΔnLF) during HD were utilized for quantification of CSR.

The study protocol was approved by the institutional review board of the En Chu Kong Hospital (ECKIRB1071202). All clinical investigations were conducted according to the principles of the Declaration of Helsinki. The prospective cohort was conducted in HD patients with chronic diseases at blood purification centers from March 2018 until February 2020. A written informed consent was

obtained from the participants of this study. Patients undergoing HD treatment for at least 3 months were eligible for inclusion. All patients had to be older than 18 years of age and receive thrice-weekly HD. Patients were excluded from the study if they had a terminal illness, active infection, active malignancy, or for personal reasons.

Emergent evidence has shown patients with impaired sympathetic response during HD are at high risk of adverse clinical outcomes. Normalized low-frequency LF (nLF) has been well-described as an index of cardiac sympathetic activity formulas:

$$nLF (\%) = LF \div (total\ power - VLF) \times 100.$$

Thus, changes in nLF (ΔnLF) before and after HD represent CSR in HD patients.

Patients undergoing hemodialysis treatment have different complications as shown in Figure 6, which can be used to observe the extent of their sympathetic and parasympathetic nerves.

Low-frequency (LF) ranges: ~0.04 Hz–0.15 Hz
High-frequency (HF) ranges: ~0.15 Hz–0.4 Hz
HF%: parasympathetic index
LF%: sympathetic index
LF/HF: sympathetic/parasympathetic balance phenomenon indicator.

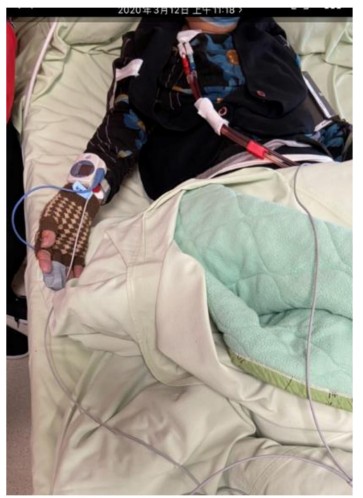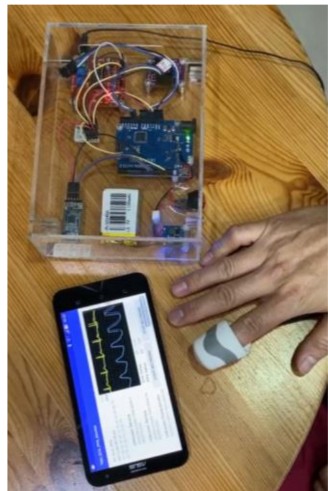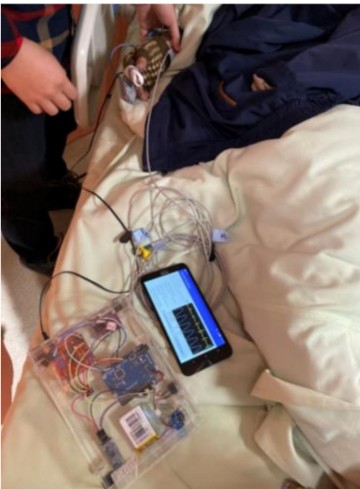

**Figure 6.** ECG/PPG continuity tracking in clinical cases.

When the supply of blood oxygen is insufficient to meet the body's metabolic rate, it is called hypoxemia. Everyone has a different tolerance for the hypoxemia condition. For example, when we do more intense exercise, our body compensates by increasing the number of breaths and increasing cardiac output, and then a faster heartbeat is necessary for compensation. In a hypoxemic state, our bodies compensate by increasing the number of breaths and increasing cardiac output, such as by increasing the heartbeat rate. If the blood oxygen level after compensation is still insufficient, symptoms will occur. Generally, patients with poor cardiopulmonary function, advanced age, and severe anemia and/or with chronic diseases such as diabetes, hypertension, and stroke, among other conditions are more prone to hypoxemia, as are renal dialysis patients, especially during dialysis. Blood oxygen saturation monitoring reflects the ability of a person's blood to transport oxygen. The metabolism of the human body is inseparable from the participation of oxygen. The human respiratory system brings oxygen into the blood, and the hemoglobin in the blood will combine with the inhaled oxygen to form a substance called oxygenated hemoglobin, which is circulated through the blood. This system is brought into the cells of various organs of the body, allowing the cells of each organ to work satisfactorily. Normal blood oxygen saturation should be in the range of 70–100%. When the simultaneous detections

of blood oxygen and heart rate are used in conjunction with a dialysis machine, the system can detect the consistency, conductivity, blood volume, albumin, and other indicators of the blood, and the current blood volume of the patient can be reversed. Once the patients' ultrafiltration was close to dry weight, the plasma refilling capacity slowed, and these monitoring values will be affected by concentrated and steeply rising blood, which is used to predict and prevent hypotension. After multiple values were gathered, the average value was stored in a patient-specific dialysis machine, providing a reference for medical staff to evaluate the amount of dehydration in future dialysis sessions.

The pulse oximeter measures blood oxygen saturation ($SpO_2$). It is the percentage of oxygenated hemoglobin that can transport oxygen in hemoglobin. The pulse rate (PR) of the arteries is measured in beats per minute. There is also the blood perfusion index (PI), which is the ratio of pulsating blood flow to non-pulsating blood flow in peripheral tissues. The higher the PI, the more accurate the $SpO_2$ measured.

The magnitude of blood pressure drop during dialysis is positively correlated with changes in body weight and blood volume before and after dialysis. Therefore, patients who gain too much weight during each dialysis period, or if medical staff underestimate the patient's dry weight, will increase the dehydration demand and exceed the load of the movement of water inside and outside the cell. This causes hypotension during dialysis. If the blood flow is redistributed, such as concentrated in the blood vessels of the gastrointestinal tract, the blood flow that can effectively return to the heart will also decrease. Therefore, eating during dialysis is also one of the causes of hypotension.

Hemodynamics caused by hemodialysis (tachycardia, peripheral perfusion) analyze the fluctuation of blood oxygen as arterial pressure (pressure difference in arteries). It is caused by the pressure difference in the blood vessels, mainly based on Ohm's law of hemodynamics. The pressure difference between any two points in the blood vessel along the blood flow direction is P (mmHg, millimeters of mercury), blood flow Q (mL/min or L/min) and peripheral resistance (PRU, peripheral resistance unit), and the three relational formula (A) is:

$$Q = \frac{P}{R'} \ (A)$$

The pressure difference in the arteries is mainly caused by the blood output of the heart and the resistance around it. These resistances include blood viscosity, blood vessel length and blood vessel radius.

The fistula does not impair the performance of the pulse rate (PR)/$SpO_2$. An arteriovenous fistula (AVF) is usually created in the patient's upper extremity through surgery to provide adequate blood flow during hemodialysis. The blood flow at the distal end of the AVF has changed, which theoretically may change the pulse oximetry ($SpO_2$) readings, systemic blood pressure, and skin temperature.

If the arteriovenous fistula is blocked and has poor function it can lead to insufficient long-term dialysis; therefore, the hemodialysis patients need to grasp the soft ball and hold the test to increase blood flow. Ball grip is a kind of blood vessel movement in the hand, and the common operation method is the natural sagging of the arm. The soft ball is held firmly for 5 s and then relaxed, and the repeated operation lasts 15 min, for at least 3–4 times a day to stimulate blood vessel expansion. The increase in degree and the diameter expansion of the blood vessels increase the blood flow of the arteriovenous fistula. Under the continuous execution of the ball grip exercise, the strength of muscle grip strength will increase, the diameter of veins and arteries will increase significantly, and the average blood flow of brachial artery, radial artery, cephalic vein, and brachial artery will also increase significantly.

$SpO_2$ analysis/pulse rate (PR) analysis also has a stable effect during hemodialysis treatment. The analysis of the HRV results for the squeeze test without the soft ball compared with the squeeze test with the soft ball are shown in Figure 7. The analysis results of 38 patients are shown in Figure 7a–c for pulse rate (PR) measurements without squeezing the soft ball. The curves of the squeeze test are without the soft ball. Patients can be the sympathetic/parasympathetic balance indicators. The analysis results of 38 patients that undertook the squeezing the soft ball test are also shown in Figure 7d–f. The results prove that the dialysis patients that squeezed soft balls have sympathetic and parasympathetic nerves,

which are usually stable and balanced. When a state of emergency (hemodialysis) is reached, such as a fear, preparation for an attack, or need to escape, the sympathetic nerves will be excited. This will cause the patients to have a rapid heartbeat, increase their blood flow, and force their breathing to become faster. After the state of emergency (hemodialysis) ends, the parasympathetic nerves begin to strengthen and function, allowing the body to rest and take care of its health, slowing the heartbeat, slowing breathing, and reducing the amount of sweating. Patients also strengthen digestion and excretion, increase absorption of nutrients, etc., to maintain basic operations of life. This belongs to a very concentrated normal distribution, as shown in Figure 7f.

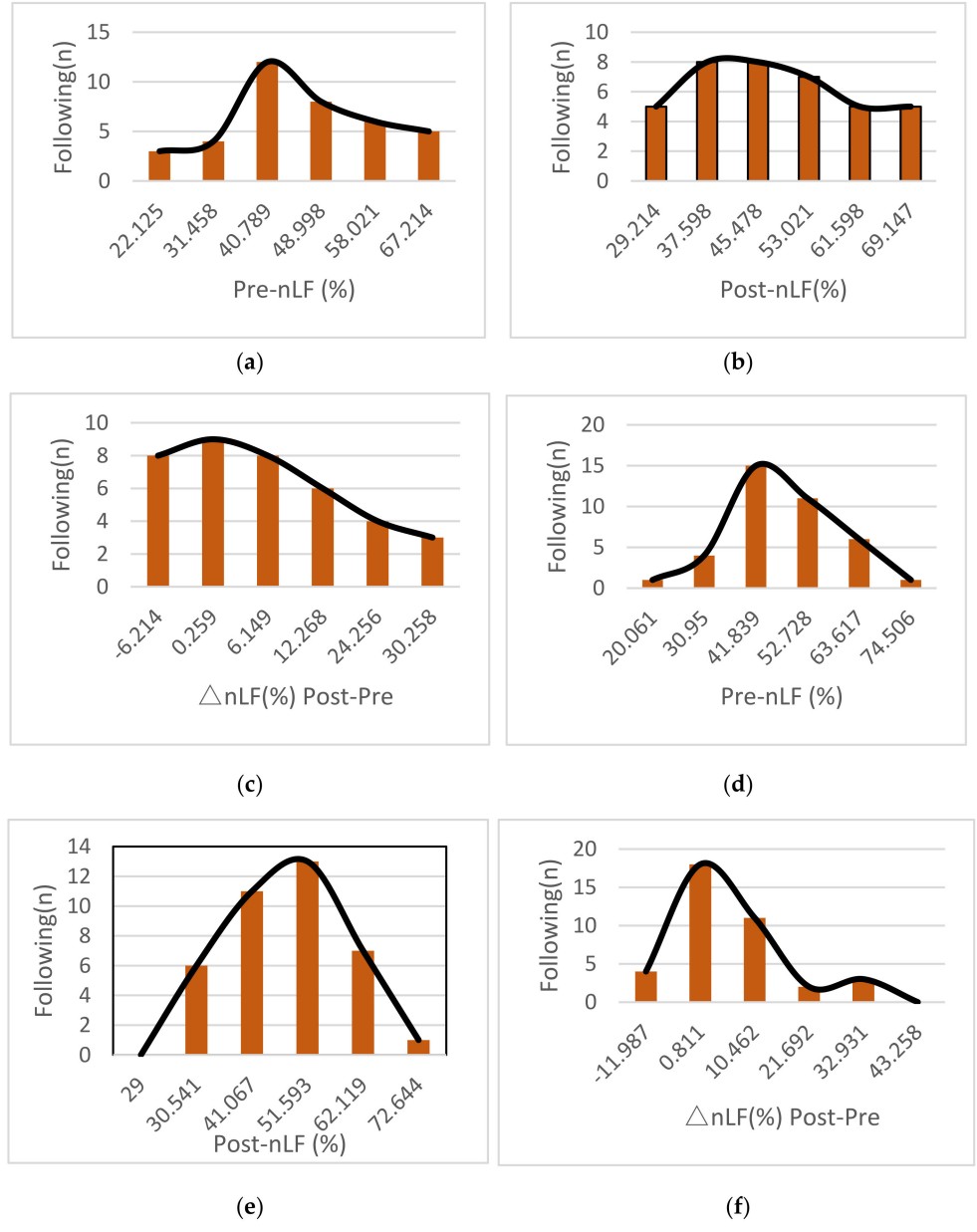

**Figure 7.** The pulse rate (PR) measurement without squeezing the soft ball (**a**) (sympathetic/parasympathetic balance indicator before-HD, (**b**) (sympathetic/parasympathetic balance indicator after hemodialysis treatment, and (**c**) autonomic nerve activation observed without squeezing the soft ball; the pulse rate (PR) measurement without squeezing the soft ball (**d**) (sympathetic/parasympathetic balance indicator before-HD, (**e**) (sympathetic/parasympathetic balance indicator after hemodialysis treatment, and (**f**) autonomic nerve activation observed with squeezing the soft ball.

Figure 7 shows analysis results through curves of 38 patients who have different complications and are undergoing hemodialysis treatment, which can be used to observe the extent of their sympathetic and parasympathetic nerves.

LF ranges: ~0.04 Hz–0.15 Hz

HF ranges: ~0.15 Hz–0.4 Hz

HF%: parasympathetic index

LF%: sympathetic index

LF/HF: sympathetic/parasympathetic balance phenomenon indicator.

Among them:

(1) 18 patients with sympathetic/parasympathetic balance indicator ((sympathetic/parasympathetic balance indicator after-HD)—(sympathetic/parasympathetic balance indicator before-HD)) value equal to 0.811.

(2) 15 patients had autonomic dysfunction, (sympathetic/parasympathetic balance indicator after-HD)—(sympathetic/parasympathetic balance indicator before-HD)) value equal to 10.462, and −11.987.

(3) Five patients who squeezed the soft ball had autonomic nervous disorders.

The equipment of the PR analyzer for the measurement of $SpO_2$ also has a stable effect during hemodialysis treatment. Hypoxemia is often an important problem for hemodialysis patients. Chronic deterioration of these patients' cardiovascular system due to hypoxemic events is often overlooked. Notably, approximately 50% of dialysis patients die from cardiovascular disease, and their risk of ischemia is five to 20 times higher than the general population. The occurrence of hypoxemia during dialysis is complex and multifactorial, due to the following reasons:

A. Pulmonary edema: renal dialysis patients often retain too much water, leading to increase in the alveolar wall edema, which makes it difficult for oxygen to enter the blood. The resultant hypoxemia gradually improves with dehydration during dialysis.

B. Biologically incompatible artificial kidneys: early fibrous membrane artificial kidneys may cause hypoxemia due to complement activation and white blood cell accumulation in pulmonary circulation during dialysis. The new-generation synthetic membrane artificial kidney is less likely to produce this phenomenon. Therefore, patients with low blood oxygen tendency should undergo synthetic membrane artificial kidney dialysis.

C. Acid–base imbalance: most patients who receive kidney dialysis have more acidic blood (pH < 7.35) before dialysis. After two hours of dialysis, the blood is exposed to alkaline dialysate. Compared with alkali (pH = 7.45), when the blood changes from acid to alkali, the affinity between oxygen molecules and heme is strengthened, making oxidized heme less likely to release oxygen to the tissues, so symptoms of hypoxia will appear.

D. Anemia: as renal dialysis patients have lower hemoglobin, their total blood oxygen content is also lower. If the pre-dialysis blood volume ratio can be increased by 33–36%, hypoxemia is less likely to occur.

E. Cardiopulmonary insufficiency: patients with heart failure, arrhythmia, or chronic obstructive pulmonary disease have lower tolerance for hypoxemia due to poor compensation functions.

F. Sleep apnea: some patients experience drowsiness during dialysis due to intermittent respiratory arrest, resulting in more severe hypoxemia.

As noted above, some patients with kidney dialysis have sleep apnea. They become lethargic during the dialysis process and experience severe hypoxia due to intermittent respiratory arrest, as shown in Figure 8. It is not easy to evaluate a patient's level of hypoxia during the dialysis process in a clinical setting. Most patients have symptoms similar to hypotension during the dialysis process, such as dizziness, hypotension, cramps, chest tightness, or arrhythmia. They may go through cyanosis or loss of circulation in the limbs. Hence, being able to measure blood oxygen content during the

dialysis process would allow hypoxemia to be detected and treated. At present, the easiest and most reliable method requires a device that detects the patients' oxyheme saturation, such as a pulse oximeter. This type of instrument has the advantages of being non-invasive and providing continuous monitoring. It operates on the principle of two lights with different wavelengths being passed through a pulsating vascular bed, and, usually, it is clamped on a finger. The difference in permeability reflects the difference between the amount of hemoglobin and oxyheme, and it is converted into oxyheme saturation and shown on a numerical display. Multiple factors can affect detection, such as insufficient blood flow at the detection location, excessive external light, excessive abnormal hemoglobin, difference in skin color, severe anemia, frequent movement of the detection site, or abnormal pulsation, especially in patients with arteriovenous fistula dialysis. Finger perfusion is poor during dialysis. When the peripheral blood vessels shrink, the resulting hypovolemia, hypotension, or hypothermia will affect the oximeter results, because of this, about 25% of dialysis patients do not yield correct data. It would be much better to continuously monitor oxygen saturation through a dialysis circuit outside the body using non-invasive optical technology, and this instrument is blood volume detection. This instrument can provide data on blood volume, hematocrit, and oxyheme saturation in real time during dialysis. As it is better to detect the oxygen hemoglobin saturation in the dialysis circuit, the similar problems with the pulsation oxygen meter will not exist and each patient can measure the correct data.

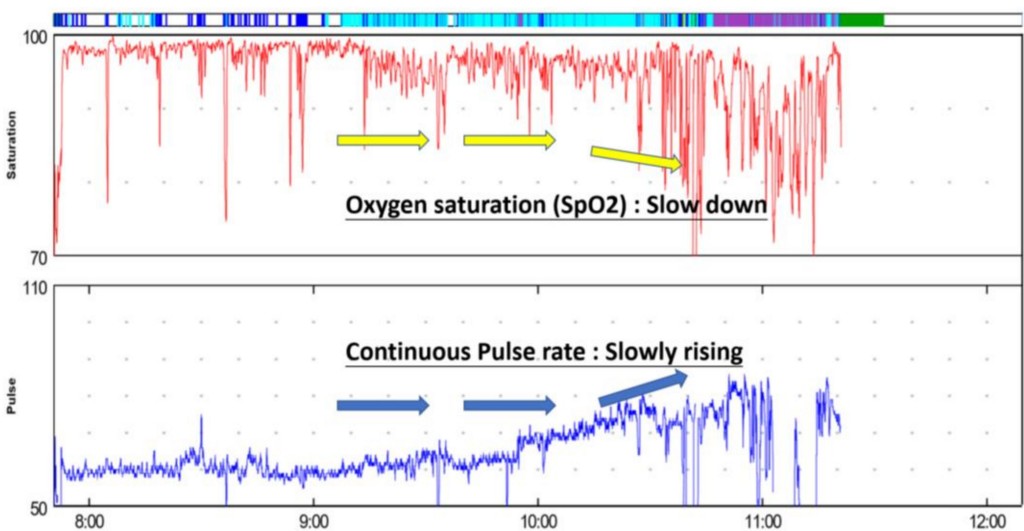

**Figure 8.** Oxygen saturation (SpO$_2$) and continuous pulse rate distribution curves during 4 h of hemodialysis treatment.

Blood oxygen saturation has a great impact on dialysis patients. When the blood oxygen saturation is less than 90%, the patients' blood volume simultaneously decreases. If the blood oxygen saturation increases, the blood volume also simultaneously increases. Rapid decline in blood volume will often be accompanied by low blood pressure and cramps, particularly in patients with blood oxygen saturation often greater than 95%. Even if the blood oxygen saturation drops, it only drops to 90%, so the impact on blood volume is not great. The current medical literature defines that if the oxygen saturation of heme is less than 90%, patients are recognized as experiencing hypoxemia [32–35]. We think that between 90% and 94% of patients are potential hypoxemia patients, where these patients experience both hypovolemia and hypoxemia saturation during dialysis. Providing oxygen in the right time can avoid discomfort.

During the 4 h hemodialysis treatment, many patients will sleep. When breathing stops, the blood oxygen level of the body drops. The brain will start a protective mechanism (brain wave awakening) because of insufficient oxygen, and then the brain quickly orders the body to wake up and interrupts sleep. This cessation of breathing will keep the sympathetic nervous system excited.

In the long run, the body is often in a state of hypoxia, which can cause inflammation and cause various diseases. This system uses the observation of the heart's rhythm change to understand the synchronization of the ECG interchange spectrum distribution. Furthermore, during the pause of breathing, we can observe the excited state of the sympathetic nervous system and whether it remains in the excited state or not. Traditionally, blood pressure records only measure once every half hour during the process of hemodialysis treatment. In this study, a comprehensive study for simultaneous detections of heart rate and blood oxygenation with high detection efficiency is shown in Figure 8. It shows the distribution curves of $SpO_2$ and continuous pulse frequency during the 4 h hemodialysis treatment. The advantage is that the results of both (heart rate and blood oxygen) directly correspond to the observation of the patient's physical state. For a patient undergoing hemodialysis treatment, there is a very urgent need for the current heart rhythm to respond to sympathetic nervous system to maintain excitement. In the same way, the downward trend of blood oxygen is detected during the 4 h of hemodialysis treatment, and the attending physician is informed.

Studies on chronic hypotension have pointed out that the relationship shows a U-shaped distribution. When the systolic blood pressure is less than 120 mm Hg, whether before or after dialysis, the mortality rate rises significantly. When we analyzed the relationship between chronic hypotension and mortality, we found that the variation still holds after excluding patients who are newly admitted or hospitalized. Chronic inflammation and comorbid cardiovascular disease have long been sufficient to explain why the chronic hypotension population has a higher mortality rate.

## 4. Conclusions

This study shows that the pulse rate (PR)/$SpO_2$ can be measured simultaneously in patients undergoing hemodialysis treatment. While a patient was undergoing 4 h dialysis, their pulse rate (PR)/$SpO_2$ were simultaneously measured. The oxygen content in the blood is directly related to the amount of heme and its saturation of oxygen. From reports of an pulse rate (PR)/$SpO_2$ equipment, the clinical patients with autonomic neuropathy were treated and evaluated. The autonomic neuropathy of 38 patients was treated and evaluated. This research shows that using a system to simultaneously observe the pulse rate (PR)/$SpO_2$ data of hemodialysis patients for 4 h could be used for microscopic examination of the patients' micro-physiological signals during the entire hemodialysis process. The results prove the "Pulse Rate measurement with squeeze the soft ball and analysis" data, and these analyzed results reveal a very concentrated normal distribution. After the state of emergency (hemodialysis) ends, the parasympathetic nerves begin to strengthen and function, allowing the body to rest and take care of its health, slowing the heartbeat, slowing breathing, and reducing the amount of sweating. Patients also strengthen digestion and excretion, increase absorption of nutrients, etc., to maintain basic operations of life. When patients with chronic hypotension undergo dialysis, clinicians should carefully evaluate whether hypotension leads to insufficient chronic dialysis and other complications.

**Author Contributions:** Methodology, J.-C.L., P.B. and C.-W.P.; formal analysis, J.-C.L. and Z.-X.C.; investigation, J.-C.L., C.-W.P. and Z.-X.C.; resources, J.-C.L. and C.-W.P.; data curation, J.-C.L., P.B., C.-W.P. and Z.-X.C.; original draft preparation, J.-C.L., P.B., C.-W.P. and Z.-X.C.; review and editing, J.-C.L. and C.-W.P. All authors have read and agreed to the published version of the manuscript.

**Funding:** This research was funded by Higher Education Sprout Project by the Ministry of Education (MOE) in Taiwan (DP2-108-21121-01-O-05-04). (DP2-109-21121-01-O-01-03, MOST-109-2918-I-038-002, TMU-NTUST-109-10, MOST-109-2221-E-038-013).

**Acknowledgments:** This work was financially supported by the Higher Education Sprout Project by the Ministry of Education (MOE) in Taiwan (DP2-108-21121-01-O-05-04). (DP2-109-21121-01-O-01-03, MOST-109-2918-I-038-002, TMU-NTUST-109-10, MOST-109-2221-E-038-013).

**Conflicts of Interest:** The authors declare no conflict of interest.

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
