# Peer review of "Autonomic Nerve Activation Observed for Hemodialysis Patients While Squeezing a Soft Ball"

_sustainability, doi:10.3390/su12229646_

Round 1

Reviewer 1 Report

Manuscript results hard to follow and not reader-friendly.

Since not extensive editing has been performed compared with the previous version of the manuscript already submitted with a final decision, I cannot suggest this manuscript for a possible publication. 

Author Response

Answer: Thanks for your comments and suggestions.

Reviewer 2 Report

The authors have improved the quality of the paper.

Author Response

(The authors gave the same response as above.)

Reviewer 3 Report

It is an interesting article. It is a well-chosen subject of this article, important clinically. After reading this article submitted to me for review, however, it occurred to me observations and comments. The described comments and suggested changes in the text lead to a better understanding of the theme and will increase readers' interest in this topic. Here they are.

1.

In the introduction, the authors omitted the parameters of renal failure determined in the clinic and did not describe new parameters used to assess renal function.

2.

In connection with the topic discussed in their research, authors should describe the role of the kidneys in the regulation of arterial blood pressure, mainly the renin-angiotensin-aldosterone axis, which will introduce readers to the scientific topic of the research.

Such a short extension of this topic will undoubtedly raise the quality of this manuscript.

Author Response

Answer: Thanks for your comments and suggestions.

1.

In the introduction, the authors omitted the parameters of renal failure determined in the clinic and did not describe new parameters used to assess renal function.

(lines 86~98.)

Renal failure parameters determined in the clinic, renal failure is a clinical condition when the kidney cells are damaged for some reason and cannot effectively remove metabolic waste and water from the body. Early renal failure (RF) (creatinine less than 5 mg/dL), the rate of hematocrit change, total CO2 (tCO2) and urea per unit change of creatinine was significantly higher than during moderate (creatinine between 5 and 10 mg/dL) or advanced (creatinine greater than 10 mg/dL) RF. If this condition is chronic and irreversible, it is called chronic renal failure. When the kidney function declines to a serious level and there are extensive systemic symptoms, it is called uremia [23].

Acute renal failure (ARF) refers to a group of clinical manifestations in which renal function declines rapidly in a short time due to the kidney itself or external factors, manifested as azotemia, water and electrolyte disorders, metabolic acidosis, anemia, and bleeding Inclination etc. Acute risk factors include volume depletion, aminoglycoside, radiocontrast exposure, septic shock, hypotension, and pigmenturia. Chronic risk factors include pre-existing renal disease, hypertension, congestive heart failure, diabetes (DM), age, liver cirrhosis.

2.

In connection with the topic discussed in their research, authors should describe the role of the kidneys in the regulation of arterial blood pressure, mainly the renin-angiotensin-aldosterone axis, which will introduce readers to the scientific topic of the research.

Such a short extension of this topic will undoubtedly raise the quality of this manuscript.

(lines 99~110.)

The role of the kidneys in the regulation of arterial blood pressure (mainly the renin-angiotensin-aldosterone axis), renin-angiotensin system (RAS) or renin-angiotensin-aldosterone system (RAAS) is a hormone system. When a lot of blood loss or blood pressure drops, this system will be activated to help regulate the body's long-term blood pressure and extracellular fluid volume (body fluid balance).

When blood pressure drops, the kidneys secrete renin. Renin catalyzes the hydrolysis of angiotensinogen to produce angiotensin I. Angiotensin I basically has no biological activity, but is cleaved by the two amino acid residues at the C-terminus by Angiotensin Converting Enzyme (ACE) to form Angiotensin II. Angiotensin II has a highly effective contraction of blood vessels, thereby increasing blood pressure; Angiotensin II can also stimulate the adrenal cortex to secrete aldosterone. Aldosterone can promote the reabsorption of water and sodium ions by the kidneys, which in turn increases fluid volume and raises blood pressure.

Round 2

Reviewer 1 Report

Despite I appreciate the aim and the relevance of the study in the field, I cannot suggest manuscript for pubblication. 

This manuscript is a resubmission of an earlier submission. The following is a list of the peer review reports and author responses from that submission.

Round 1

Reviewer 1 Report

This study shows that PR (Pulse Rate) / SPO2 can be measured simultaneously in patients undergoing hemodialysis treatment. This ratio showed that micro-physiological signals were captured during the entire hemodialysis process.

Major revision:

-The aim of the study is not clear. Please, underline it in the introduction

-Data about dialysis modalities (HD, HDF, CVVH) and dialysis vintage is not reported.

-limitation of  PR (Pulse Rate) / SPO2  during hemodynamic changes induced by hemodialysis (tachycardia, peripheral perfusion)

-can fistula impair the performance of PR (Pulse Rate) / SPO2 ? 

Author Response

Thanks for comments and suggestions. We have modified it.

1.-The aim of the study is not clear. Please, underline it in the introduction (Page 2)

2.-Data about dialysis modalities (HD, HDF, CVVH) and dialysis vintage is not reported. (Page 6-7)

3.-limitation of  PR (Pulse Rate) / SPO2  during hemodynamic changes induced by hemodialysis (tachycardia, peripheral perfusion) (Page 12)

4.-can fistula impair the performance of PR (Pulse Rate) / SPO2 ?  (Page 12)

========

1.-The aim of the study is not clear. Please, underline it in the introduction (Page 2)

Answer:

  1. Introduction

The aim of the study is using a system for simultaneously observing the PR (Pulse Rate) / SPO2 data of hemodialysis patients for 4-h could be used for microscopic examination of the patients’ micro-physiological signals during the entire hemodialysis process. Cardiac sympathetic response (CSR) and malnutrition-inflammation syndrome (MIS) with squeeze test with the soft ball score are validated assessment tools for patients’ health condition. We aim to evaluate the joint effect of CSR and MIS on all-cause and cardiovascular (CV) mortality in patients with hemodialysis (HD). The Methods, SPO2 analysis/Pulse Rate (PR) analysis also has a stable effect during hemodialysis treatment. Changes in normalized low frequency (ΔnLF) during HD were utilized for quantification of CSR.

  1. -Data about dialysis modalities (HD, HDF, CVVH) and dialysis vintage is not reported. (Page 6-7)

Answer:

Hemodialysis (HD) is through the diffusion function (Diffusion), the toxins are discharged through the concentration difference, and the filtration volume is <0.5-1.0 L / hour. The molecules that can be removed are smaller. The molecular weight of uremic molecules removed by hemodialysis is about 1000 or less. Hemofiltration (Hemofiltration) gives the blood a Positive Hydrostatic Pressure. The dialysate produces Negative Hydrostatic Pressure and convection (Convection), and toxins are filtered out through a semi-permeable membrane. The filtration capacity of hemofiltration is> 1.0-2.0 L/hour. The filterable molecular weight can reach 25,000, but the small molecules are not clean. It can also be described that hemodiafiltration has both the diffusion effect of hemodialysis (good removal efficiency for small molecules) and the convection effect of blood filtration (a certain removal rate for large, medium and small molecules). It can be said to be a treatment that combines the best of both. Compared with hemodialysis (HD), HDF can increase the clearance of small molecules, and for medium and large molecules that cannot be removed by normal hemodialysis. The clearance rate is n times. It can be used to prevent and ameliorate diseases caused by the deposition of large molecules such as amyloidosis or urotoxic neuropathy.

Continuous venovenous hemofiltration (CVVH), CVVH, double-lumen catheter is inserted in the vein. Use the pump to pull the blood out through the dialyzer and then return to the body to remove excess water and toxins. It is to put a double-lumen catheter into the large vein and slowly remove excess water and urinary toxins from the body with a low blood flow rate (about 150-200ml/minute) without interruption for 24 hours. Because Conventional Hemodialysis cannot meet the medical needs, the performance of HD is only equivalent to that of Native Kidney in stage 5 chronic kidney disease. In addition, hemodialysis is not effective in removing medium and large molecules of urinary toxins, and complications such as hypotension often occur during the process. In addition, the purity of traditional dialysate is insufficient, and inflammation is prone to occur. Therefore, there is continuous venous hemofiltration (CVVH). Regarding dialysis methods (HD, HDF, CVVH), the indications of treatment are different, the principles are slightly different, and the treatment time is also different. The more severe the patient's condition, the more CVVH is used. In CVVH mode, dialysis is performed for 24 hours (-2.0kg/24hours). For HD or HDF mode, it is -2.0kg/4hours.

Hemodialysis methods (HD, HDF, CVVH)

Characteristics and dialysis time

HemodialysisB(HD)

1.HD performance is only equivalent to the Native Kidnet of Stage 5 CKD.

2.HD is not effective in removing medium and large molecules of urinary toxins.

3.Complications such as hypotension often occur during HD.

4.Filter capacity< 0.5-1.0 L / hour.

Hemofiltration

1.The MW of blood filtration can reach 25000, but the small molecules are not clean.

2.Better cardiovascular stability, less prone to hypotension.

3.Filter capacity> 1.0-2.0 L / hour.

Hemodia-filtration (HDF)

1.Less blood pressure drop.

2.Could reduce musculoskeletal symptoms caused by long-term dialysis treatment.

3.Improve nerve conduction velocity.

4.Good macromolecular uremic (ß2-Microglobulin) clearance rate.

5.Good small molecule clearance rate (URR).

6.Hemodiafiltration (HDF) = Hemodialysis (HD) + Hemofiltration (HF)

CVVH (continuous venovenous hemofiltration (CVVH))

1.24 hours without interruption, low blood flow rate (about 150-200ml/minute),

2.Slowly remove excess water and urinary toxins from the body.

3.In CVVH, the double-lumen catheter is inserted into the vein, and the blood is pulled out through the dialyzer and returned to the body by the pump to achieve the purpose of removing excess water and toxins.

HD vintage was defined as the duration of time between the first day of HD treatment and the first day that the patient entered the study cohort. Each HD session was performed for 3.5–4.5 h with a blood flow rate of 200–300 mL/min and dialysate flow rate of 500 mL/min. Blood pressure was recorded in the horizontal recumbent position before dialysis session. Pre-dialysis blood samples were obtained from the existing vascular access.

3.

-limitation of  PR (Pulse Rate) / SPO2  during hemodynamic changes induced by hemodialysis (tachycardia, peripheral perfusion) (Page 12)

Answer:

The pulse oximeter measures blood oxygen saturation (SpO2). It is the percentage of oxygenated hemoglobin that can transport oxygen in hemoglobin. The pulse rate (PR) of the arteries. It is measured in beats per minute; there is also blood perfusion index (PI). It is the ratio of pulsating blood flow to non-pulsating blood flow in peripheral tissues. The higher the PI, the more accurate the SpO2 measured.

The magnitude of blood pressure drop during dialysis is positively correlated with changes in body weight and blood volume before and after dialysis. Therefore, patients who gain too much weight during each dialysis period, or medical staff underestimate the patient's dry weight, will increase the dehydration demand and exceed the load of the movement of water inside and outside the cell. This causes hypotension during dialysis. If the blood flow is redistributed, such as concentrated in the blood vessels of the gastrointestinal tract, the blood flow that can effectively return to the heart will also decrease. Therefore, eating during dialysis is also one of the causes of hypotension.

Hemodynamics caused by hemodialysis (tachycardia, peripheral perfusion). It analyzes the fluctuation of blood oxygen as arterial pressure (pressure difference in arteries). It is caused by the pressure difference in the blood vessels. Mainly based on Ohm's law in hemodynamics. The pressure difference between any two points in the blood vessel along the blood flow direction is P (mmHg, millimeters of mercury), blood flow Q (ml/min or L/min) and peripheral resistance (PRU, peripheral resistance unit), and the three The relational formula is as (A) formula:

The pressure difference in the arteries is mainly caused by the blood output of the heart and the resistance around it. These resistances include blood viscosity, blood vessel length and blood vessel radius.

4.-can fistula impair the performance of PR (Pulse Rate) / SPO2 ? (Page 13)

Answer:

The fistula does not impair the performance of PR (pulse rate)/SPO2.Arteriovenous fistula (AVF) is usually created in the patient's upper extremity through surgery to provide adequate blood flow during hemodialysis. The blood flow at the distal end of the AVF has changed, which theoretically may change the pulse oximetry (SpO2) readings, systemic blood pressure, and skin temperature.

Reviewer 2 Report

In the manuscript by Jian-Chiun Liou et al., the authors showed ECG and PPG can be detected simultaneously to observe physiological phenomena in patients undergoing hemodialysis. The aim of the study is really interesting despite it is not clear to the reader if authors have created a novel biosensor or if they have implemented the use of what is already available nowadays.

The manuscript appears hard to follow. No method section is present.  

The introduction is not well structured. It should introduce to the reader what is the problem the manuscript is intended to face up. Then clearly defined the aim of the study. Instead, now the introduction appears confusing. No description of what is haemodialysis, how it works and why people have to attend this kind of therapy. Neither the percentage of subjects treated every day with hemodialysis.  

Does Figure 1 and Figure 2 represent the work performed by the authors? If it is so, they should state it in the text.

Table 1 is unacceptable. Patients are not items and disease or compliance are not problems. What does the author mean with Poorly controlled hypertension or detour? Are they clinical diagnosis or have been arbitrarily decided by the authors? Patients recruitment section and clinical diagnostic criteria have to be reported in the manuscript (method section). In which hospital this analysis has been performed? Ethical committee approval is available?

Authors reported gender as a % but the only ratio is present. Adjust it.

Further, if 38 patients have been analyzed I expect a summary table with all the results. Authors should provide it.

Formulas have to be inserted correctly in the text.

No literature has been cited. Neither when authors reported in line 383: “The current medical literature defines that as the saturation oxygen of  heme is less than 90%, patients are recognized as hypoxemia”.

I suggest authors should provide an extensive editing of the manuscript by tacking advantage of clinical help . This can for sure be useful to the work.

Despite the manuscript showed interesting results, I cannot suggest this manuscript in this form for a possible publication

Author Response

Thanks for comments and suggestions. We have modified it.

1.The manuscript appears hard to follow. No method section is present.  (Page 2)

2.The introduction is not well structured. It should introduce to the reader what is the problem the manuscript is intended to face up. Then clearly defined the aim of the study. Instead, now the introduction appears confusing. No description of what is haemodialysis, how it works and why people have to attend this kind of therapy. Neither the percentage of subjects treated every day with hemodialysis.  (Page 2-3)

3.Does Figure 1 and Figure 2 represent the work performed by the authors? If it is so, they should state it in the text. (Page 4-5)

4.Table 1 is unacceptable. Patients are not items and disease or compliance are not problems. What does the author mean with Poorly controlled hypertension or detour? Are they clinical diagnosis or have been arbitrarily decided by the authors? Patients recruitment section and clinical diagnostic criteria have to be reported in the manuscript (method section). In which hospital this analysis has been performed? Ethical committee approval is available? (Page 9-10)

5.Authors reported gender as a % but the only ratio is present. Adjust it. (Page 10)

6.Further, if 38 patients have been analyzed I expect a summary table with all the results. Authors should provide it. (Page 10)

7.Formulas have to be inserted correctly in the text. (Page 10)

8.No literature has been cited. Neither when authors reported in line 383: “The current medical literature defines that as the saturation oxygen of  heme is less than 90%, patients are recognized as hypoxemia”. (Page 14, Page 17)

I suggest authors should provide an extensive editing of the manuscript by tacking advantage of clinical help . This can for sure be useful to the work.

Despite the manuscript showed interesting results, I cannot suggest this manuscript in this form for a possible publication

Answer:

Thanks for comments and suggestions. We have modified it.

============

1.The manuscript appears hard to follow. No method section is present.  (Page 2)

Answer:

The aim of the study is using a system for simultaneously observing the PR (Pulse Rate) / SPO2 data of hemodialysis patients for 4-h could be used for microscopic examination of the patients’ micro-physiological signals during the entire hemodialysis process. Cardiac sympathetic response (CSR) and malnutrition-inflammation syndrome (MIS) with squeeze test with the soft ball score are validated assessment tools for patients’ health condition. We aim to evaluate the joint effect of CSR and MIS on all-cause and cardiovascular (CV) mortality in patients with hemodialysis (HD). The Methods, SPO2 analysis/Pulse Rate (PR) analysis also has a stable effect during hemodialysis treatment. Changes in normalized low frequency (ΔnLF) during HD were utilized for quantification of CSR.

2.The introduction is not well structured. It should introduce to the reader what is the problem the manuscript is intended to face up. Then clearly defined the aim of the study. Instead, now the introduction appears confusing. No description of what is haemodialysis, how it works and why people have to attend this kind of therapy. Neither the percentage of subjects treated every day with hemodialysis.  (Page 2-3)

Answer:

The kidneys can metabolize substances harmful to the human body and play an important role in maintaining life. If you suffer from diabetes, kidney stones, renal arteriosclerosis and other diseases and not properly treated, the kidney is damaged for more than three months, it will cause chronic kidney disease, and then damage the kidney function. Therefore, such patients must receive hemodialysis treatment.

Hemodialysis involves sticking one end of a puncture needle into the distal end of the arteriovenous fistula. It drains the blood out of the body, removes urinary toxins and water through the diffusion and ultrafiltration of the artificial kidney (kidney dialysis machine) semipermeable membrane, and then leads the blood back to the proximal end of the arteriovenous fistula. The duration of a kidney dialysis is about 4 hours, and the treatment is 3 times a week.

Hemodialysis is through the diffusion function (Diffusion), so that toxins are discharged through concentration differences. The filtration capacity is less than 0.5-1.0 L/hour, and the molecules that can be removed are small. The molecular weight of uremic molecules removed by hemodialysis is about 1000 or less. Hemofiltration gives the blood a Positive Hydrostatic Pressure, the dialysate produces Negative Hydrostatic Pressure and convection (Convection), and toxins are filtered out through a semipermeable membrane. The filtration volume of Hemofiltration> 1.0-2.0 L / hour, the filterable molecular weight can reach 25000, but the small molecules are not clean. The percentage of subjects receiving hemodialysis treatment every day is 1/5.

3.Does Figure 1 and Figure 2 represent the work performed by the authors? If it is so, they should state it in the text. (Page 4-5)

Answer:

The work performed of main research experimental focus:

(1) For Hemodialysis Patients, the high-precision SpO2 value and pulse rate architectures are the solutions combining analog front-ends and analog-to-digital converter singles (ADC).

(2) For patients undergoing hemodialysis, clinically capture the physiologically analysis signals of SPO2 from PR Analyzer by external heart rate acquisition system.

(3) For mortality risk assessment of hemodialysis patients, the observation of autonomic nerve activation, which analyzes and collects the signals of HRV after hemodialysis treatment.

(4) The next stage is to obtain SPO2 signal from PR Analyzer. These patients undergoing hemodialysis treatment have different complications. For this reason, with the progress of hemodialysis treatment, observations on the degree of influence of HRV on sympathetic and parasympathetic nerves are also underway.

(5) ECG and PPG can be detected simultaneously to observe physiological phenomena in patients undergoing hemodialysis. The ECG / PPG signals can be synchronous sampling by using ADC, and synchronous counting architecture simultaneously drives sampling blood oxygen and heart rate detection to achieve the applications of both continuous detections.

(6) SPO2 analysis/Pulse Rate (PR) analysis also has a stable effect during hemodialysis treatment. Changes in normalized low frequency (ΔnLF) during HD were utilized for quantification of CSR.

4.Table 1 is unacceptable. Patients are not items and disease or compliance are not problems. What does the author mean with Poorly controlled hypertension or detour? Are they clinical diagnosis or have been arbitrarily decided by the authors? Patients recruitment section and clinical diagnostic criteria have to be reported in the manuscript (method section). In which hospital this analysis has been performed? Ethical committee approval is available? (Page 9-10)

Answer :

4.1

Table 1 is unacceptable. Patients are not items and disease or compliance are not problems. What does the author mean with Poorly controlled hypertension or detour? Are they clinical diagnosis or have been arbitrarily decided by the authors? Patients recruitment section and clinical diagnostic criteria have to be reported in the manuscript (method section). In which hospital this analysis has been performed? Ethical committee approval is available? (Page 11-12)

Answer:

Poorly controlled hypertension, which is responsible for occurrence of stroke. This means patients who also have stroke symptoms. An aortic aneurysm is an enlargement (dilatation) of the aorta to greater than 1.5 times normal size. They usually cause no symptoms except when ruptured. Occasionally, there may be abdominal, back, or leg pain. The blood vessel of bypass surgery, Bypass surgery can be used for patients with multiple stenosis or blockage of the blood vessel. The above patients with these symptoms are clinically diagnosed.

4.2

Table 1 is unacceptable. Patients are not items and disease or compliance are not problems. What does the author mean with Poorly controlled hypertension or detour? Are they clinical diagnosis or have been arbitrarily decided by the authors? Patients recruitment section and clinical diagnostic criteria have to be reported in the manuscript (method section). In which hospital this analysis has been performed? Ethical committee approval is available?

Answer:

Patients with chronic diseases recruitment section and clinical diagnostic criteria background, the aim of the study is using a system for simultaneously observing the PR (Pulse Rate) / SPO2 data of hemodialysis patients for 4-h could be used for microscopic examination of the patients’ micro-physiological signals during the entire hemodialysis process. Cardiac sympathetic response (CSR) and malnutrition-inflammation syndrome (MIS) with squeeze test with the soft ball score are validated assessment tools for patients’ health condition. We aim to evaluate the joint effect of CSR and MIS on all-cause and cardiovascular (CV) mortality in patients with hemodialysis (HD). The Methods, SPO2 analysis/Pulse Rate (PR) analysis also has a stable effect during hemodialysis treatment. Changes in normalized low frequency (ΔnLF) during HD were utilized for quantification of CSR.

For patients’ health condition. the extent of their sympathetic and parasympathetic nerves.

LF ranges : 0.04 Hz ~ 0.15 Hz.

HF ranges : 0.15 Hz ~ 0.4 Hz.

HF% : parasympathetic index.

LF% : sympathetic index

LF/HF : sympathetic/parasympathetic balance phenomenon indicator.

4.3

Table 1 is unacceptable. Patients are not items and disease or compliance are not problems. What does the author mean with Poorly controlled hypertension or detour? Are they clinical diagnosis or have been arbitrarily decided by the authors? Patients recruitment section and clinical diagnostic criteria have to be reported in the manuscript (method section). In which hospital this analysis has been performed? Ethical committee approval is available?

Answer:

The study protocol was approved by the Institutional Review Board of the En Chu Kong Hospital (ECKIRB1071202). All clinical investigations were conducted according to the principles of the Declaration of Helsinki. The prospective cohort was conducted in HD patients with chronic diseases at blood purification centers from March 2018 until February 2020. A written informed consent was obtained from the participants of this study. Treatment of chronic diseases, patients undergoing HD treatment for at least 3 months were eligible for inclusion. All patients had to be older than 18 years of age and receive thrice-weekly HD. Patients were excluded from the study if they had terminal illness, active infections, active malignancy, or personal reasons.

5.Authors reported gender as a % but the only ratio is present. Adjust it. (Page 10)

Answer:

Items

Patients (n = 38)

%

Age, years, mean ± SD (hemodialysis treatment)

68 ± 12

Gender, n (%)

Male (hemodialysis treatment)

18 / 38

47.4%

Female (hemodialysis treatment)

20 / 38

52.6%

ΔnLF (%) with sympathetic/parasympathetic balance indicator (value equal to 0.811)

18 / 38

47.4%

ΔnLF (%) with autonomic dysfunction positive (value equal to 10.462)

11 / 38

28.9%

ΔnLF (%) with autonomic dysfunction negative (value equal to -11.987)

4 / 38

10.5%

ΔnLF (%) with autonomic nervous disorders.

5/ 38

13.2%

Problem issue:

With Poorly controlled hypertension ratio

32 / 38

84.2%

With Aortic aneurysm ratio

16 / 38

42.1%

With The blood vessel of bypass surgery ratio

28 / 38

73.7%

The squeeze test with the soft ball

38 / 38

100%

Emergent evidences have shown patients with impaired sympathetic response during HD are at high risk of adverse clinical outcomes. Normalized LF (nLF) has been well-described as an index of cardiac sympathetic activity Formulas: nLF (%) = LF/(total power-VLF)∗ 100. Thus changes in nLF (ΔnLF) before and after HD represent CSR in HD patients.

6.Further, if 38 patients have been analyzed I expect a summary table with all the results. Authors should provide it. (Page 10)

Answer:

Items

Patients (n = 38)

%

Age, years, mean ± SD (hemodialysis treatment)

68 ± 12

Gender, n (%)

Male (hemodialysis treatment)

18 / 38

47.4%

Female (hemodialysis treatment)

20 / 38

52.6%

ΔnLF (%) with sympathetic/parasympathetic balance indicator (value equal to 0.811)

18 / 38

47.4%

ΔnLF (%) with autonomic dysfunction positive (value equal to 10.462)

11 / 38

28.9%

ΔnLF (%) with autonomic dysfunction negative (value equal to -11.987)

4 / 38

10.5%

ΔnLF (%) with autonomic nervous disorders.

5/ 38

13.2%

Problem issue:

With Poorly controlled hypertension ratio

32 / 38

84.2%

With Aortic aneurysm ratio

16 / 38

42.1%

With The blood vessel of bypass surgery ratio

28 / 38

73.7%

The squeeze test with the soft ball

38 / 38

100%

Emergent evidences have shown patients with impaired sympathetic response during HD are at high risk of adverse clinical outcomes. Normalized LF (nLF) has been well-described as an index of cardiac sympathetic activity Formulas: nLF (%) = LF/(total power-VLF)∗ 100. Thus changes in nLF (ΔnLF) before and after HD represent CSR in HD patients.

7.Formulas have to be inserted correctly in the text. (Page 10)

Answer:

Emergent evidences have shown patients with impaired sympathetic response during HD are at high risk of adverse clinical outcomes. Normalized LF (nLF) has been well-described as an index of cardiac sympathetic activity Formulas: nLF (%) = LF/(total power-VLF)∗ 100. Thus changes in nLF (ΔnLF) before and after HD represent CSR in HD patients.

Patients in curves, who have different complications undergoing hemodialysis treatment, which can be used to observe for that the extent of their sympathetic and parasympathetic nerves.

LF ranges : 0.04 Hz ~ 0.15 Hz.

HF ranges : 0.15 Hz ~ 0.4 Hz.

HF% : parasympathetic index.

LF% : sympathetic index

LF/HF : sympathetic/parasympathetic balance phenomenon indicator

8.No literature has been cited. Neither when authors reported in line 383: “The current medical literature defines that as the saturation oxygen of  heme is less than 90%, patients are recognized as hypoxemia”. (Page 14, Page 17)

I suggest authors should provide an extensive editing of the manuscript by tacking advantage of clinical help . This can for sure be useful to the work.

Despite the manuscript showed interesting results, I cannot suggest this manuscript in this form for a possible publication

Answer:

Blood oxygen saturation has a great impact on dialysis patients. When the blood oxygen saturation is less than 90%, the patients’ blood volume simultaneously decreases. If the blood oxygen saturation increases, the blood volume also simultaneously increases. Rapid decline in blood volume will often be accompanied by low blood pressure and cramps, patients with blood oxygen saturation often greater than 95%. Even if the blood oxygen saturation drops, it only drops to 90%, so the impact on blood volume is not great. The current medical literature defines that as the saturation oxygen of heme is less than 90%, patients are recognized as hypoxemia [32-35]. We think that between 90% and 94% of patients are potential hypoxemia patients, when these patients have both hypovolemia and hypoxemia saturation during dialysis. At the right time, giving oxygen in time can avoid the discomfort.

Reference :

  1. Gift, A.G., Stanik,J., Karpenick, J., Whitmore, K., Bolgiano, C. S. Oxygen Saturation in Postoperative Patients at Low Risk for Hypoxemia: Is Oxygen Therapy Needed?. Anesthesia & Analgesia, vol. 80, no. 2 , 368-372, 1995.
  2. Setty, B.N., Stuart, M.J., Dampier, C. Hypoxaemia in sickle cell disease: biomarker modulation and relevance to pathophysiology. Lancet, vol. 362, no. 2, pp. 1450-1455, 2003.
  3. Rackoff, W.R., Kunkel, N. Silber, J.H. Asakura, T., Ohene-Frempong, K. Pulse oximetry and factors associated with hemoglobin oxygen desaturation in children with sickle cell disease. Blood, 81, 3422-3427, 1993.
  4. Homi, J., Levee, L., Higgs, D., Thomas, P., Serjeant, G. Pulse oximetry in a cohort study of sickle cell disease. Clin Lab Haematol, 19 , 17-22, 1997.

I suggest authors should provide an extensive editing of the manuscript by tacking advantage of clinical help . This can for sure be useful to the work.

Despite the manuscript showed interesting results, I cannot suggest this manuscript in this form for a possible publication

Answer:

Thanks for comments and suggestions. We have modified it.

Round 2

Reviewer 2 Report

In this revised version and/or in their rebuttal letter, authors meet all comments and critics raised by reviewers. I consider the revised version of the manuscript as highly improved but I still recommend authors to modify as follow:

-authors widelly use to soubdivide each points as : 1;2;3;.. please modify not to use so often.

-table 1: if % is already repported as such for example in "gender (%) ", it is redundant to add an additional column. all the % in table 1 can be addedd in the same column where patients informations are provided. 

-sosbtitute "items" with  "demographic and clinical infromations".

-does for author "problem issue" means "disease"? if it is so I don't think the manuscript has been revised by a clinic. 

Despite I appreciate the authors provide some of the corrections requested, I still think the manuscript is not still acceptable for a pubblication.